# Regional Prevalence of Dyslipidemia, Healthcare Utilization, and Cardiovascular Disease Risk in South Korean: A Retrospective Cohort Study

**DOI:** 10.3390/ijerph18020538

**Published:** 2021-01-11

**Authors:** Kyu-Tae Han, SeungJu Kim

**Affiliations:** 1Division of Cancer Control and Policy, National Cancer Center, Goyang 10408, Korea; kthan.phd@gmail.com; 2Department of Nursing, College of Nursing, Eulji University, Seongnam 13135, Korea

**Keywords:** dyslipidemia, cardiovascular disease, health expenditure, health disparities, regional disparities

## Abstract

Background: Health disparities between different populations have long been recognized as a problem, and they are still an unsolved public health issue. Many factors can make a difference, and disparities for cardiovascular diseases (CVDs) are especially pronounced. This study aimed to assess South Korean regional variations for dyslipidemia prevalence, differences in healthcare utilization, and CVD risk. Methods: We used data from 52,377 patients from the National Health Insurance Sampling. Outcome variables were the risk of CVD, healthcare utilization (outpatient visits), and healthcare expenditures. A generalized estimating equation model was used to identify associations between the region and CVD risk, a Poisson regression model was used for evaluating outpatient visits, and a generalized linear model (gamma and log link function) was used to evaluate healthcare expenditures. Results: A total of 12,443 (23.8%) patients were diagnosed with CVD. Dyslipidemia prevalence varied by region, and the most frequent dyslipidemia factor was high total cholesterol. CVD risk was increased in low population-density regions compared to high-density regions (odds ratio [OR]: 1.133, 95% confidence interval [CI]: 1.037–1.238). Healthcare expenditures and outpatient visits were also higher in low-density regions compared to high-density regions. Conclusions: This study provides a regional assessment of dyslipidemia prevalence, healthcare utilization, and CVD risk. To bridge differences across regions, consideration should be given not only to general socio-economic factors but also to specific regional factors that can affect these differences, and a region-based approach should be considered for reducing disparities in general health and healthcare quality.

## 1. Introduction

Cardiovascular disease (CVD) is the leading cause of death worldwide, and many factors have been identified as targets for reducing its prevalence. Most population CVD risk factors, such as high total cholesterol levels and hypertension, are modifiable and can be reduced by changes in behavior [1,2]. In other words, changes in healthcare and the adoption of healthy behaviors, including physical activity and dietary habits, in the general population mean that people can live healthier lives. However, some populations are more vulnerable than others due to health and healthcare disparities, and these health gaps between different populations have been recognized as an important issue to be addressed [3].

Health disparities between populations have long been recognized as a problem, and they are still an unsolved public health issue. Many factors, including race/ethnicity, urban versus rural location, and socio-economic status influence healthcare, especially CVD [4,5,6]. CVD prevalence varies geographically from region to region, and patients have poor outcomes in areas where disease management is difficult [7]. Access to healthcare is an important factor that contributes to the widening health gap between urban and rural areas, leading to further CVD risk-factor disparities [8]. Residents living in urban areas have more opportunities to visit healthcare services than in rural areas, resulting in lower mortality and morbidity and exacerbating health disparities between regions [9]. Several interventions have been applied to reduce this rural versus urban health gap, and some have had positive results in reducing blood pressure and blood cholesterol levels [10]. However, these efforts have not reduced these health disparities, and this problem will only become more severe, especially in patients with chronic diseases.

Dyslipidemia is one CVD risk factor, and its prevalence is increasing with lifestyle changes. In Korea, the prevalence of dyslipidemia was 16.58% in 2013, but only 24.14% of patients were aware of their condition, and the treatment rate was low [11,12]. In other words, dyslipidemia is considered less important than other chronic diseases such as hypertension and diabetes and therefore may affect a patient’s disease management and outcome. Furthermore, this difference in emphasis can be different in rural versus urban areas, and if patients are not properly managed, it can lead to even wider health gaps.

In the Asian population, previous studies have shown that the prevalence of dyslipidemia is higher in urban than in rural areas [13,14], that obesity prevalence shows a similar pattern, and that CVD risk factors are different in rural compared to urban areas [15]. For stroke patients, there are large regional differences in healthcare quality; to reduce these, changes to better maintain continuity of care through physician allocation efficiency have been suggested [16,17]. In Korean high-risk groups for CVD, the need for effective strategies to better control low-density lipoprotein cholesterol (LDL-C) levels has also been identified [18]. Although many previous studies have evaluated dyslipidemia prevalence by region, none have examined regional dyslipidemia patient outcomes and healthcare utilization.

This study aimed to assess regional dyslipidemia prevalence, differences in healthcare utilization by dyslipidemia patients, and their CVD risk. For healthcare utilization, we assessed the number of outpatient visits and healthcare expenditures.

## 2. Materials and Methods

### 2.1. Database and Data Collection

This study used the National Health Insurance Sampling (NHIS) cohort data from 2007 to 2015. The baseline population, 1,025,340 participants who were randomly selected, represented 2.2% of the total eligible Korean population in 2002 [19]. These data included personal demographic information, medical treatment data, health examinations, and hospital characteristics. Health examinations occurred biennially or annually according to workplace rules; blue-collar workers had examinations annually. Medical data for all subjects were included as part of insurance-claim data and included diagnoses, comorbidities, medications, visit dates, and costs. In addition, we obtained regional population data from Statistics Korea based on the smallest administrative unit available (*si-gun-gu*) in Korea.

We defined newly diagnosed dyslipidemia based on International Classification of Disease (ICD)-10 codes (E78) and based on patients who were prescribed statin medications. A total of 171,750 patients were newly diagnosed with dyslipidemia from 2007 to 2014. Exclusion criteria were: patients diagnosed with dyslipidemia between 2007 to 2008; patients diagnosed either in long-term care facilities or in hospitals; patients under 20 years old; patients without health examinations; patients without serum cholesterol information; patients diagnosed with CVD before being diagnosed with dyslipidemia, and patients with incomplete demographic or health examination information. After these exclusions, 52,377 patients were included in the study.

### 2.2. Variables

Based on Statistics Korea population data from the original 257 *si-gun-gu* administrative regions, we created six population categories: <100,000; 100,000–200,000; 200,000–300,000; 300,000–400,000; 400,000–500,000, and >500,000. In general, population increased with more development or with proximity to a metropolitan area. The *si-gun-gu* populations in Seoul, the capital of South Korea, varied from 200,000 to over 500,000, but those from Gangwon-do (a rural area) had less than 100,000 people.

Outcome variables included healthcare utilization and CVD risk for patients with dyslipidemia. Healthcare utilization included the average number of outpatient visits per year and total annual healthcare expenditures during the study period. Outpatient visits were counted based on the main diagnosis indicated by insurance-claim data (ICD-10 code E78), and healthcare expenditures included inpatient and outpatient care except for medical costs not covered by National Health Insurance. Only visits and costs for dyslipidemia treatment were included. CVD was an assessment based on ICD-10 codes and included IHD (I20–I25), HTN (I10–I15), and cerebrovascular disease (I60–I69). During the study periods, those diagnosed with the ICD-10 codes as the main diagnosis was considered to develop CVD. Data for serum cholesterol levels included total cholesterol (TC), triglyceride (TG), high-density lipoprotein cholesterol (HDL-C), and low-density lipoprotein cholesterol (LDL-C) to evaluate regional dyslipidemia patient prevalence. Cholesterol levels were defined according to 2018 Korean Dyslipidemia Management guidelines [12] as follows: high TC, ≥240 mg/dL; high TG, ≥200 mg/dL; high LDL-C, ≥160 mg/dL; low HDL-C, <40 mg/dL. Based on these serum cholesterol levels, the distribution of abnormal serum cholesterol levels by region in Korea was evaluated.

Diabetes diagnosis was measured based on the ICD10 code (E10–E14). Medication data included whether or not the patient was prescribed a statin at diagnosis. To consider patients with a high risk of CVD among dyslipidemia patients, we scored the major risk factors for CVD and considered them as variables. The risk score was calculated based on age (male, ≥45 years; female, ≥55 years), positive family history of coronary artery disease, hypertension (systolic blood pressure [BP], ≥140 mmHg or diastolic BP, ≥90 mmHg), positive history of smoking, and low HDL-C (<40 mg/dL) [12]. Patients with high HDL-C (≥6 0mg/dL) are considered protective factors and are coded as –1. For each risk factor, patients were coded 1 or 0 (except high HDL-C), and the final scores were summed and categorized as 0, 1, 2, or ≥3. The data were adjusted for demographic characteristics by sex (male, female), age, income (low, low-moderate, moderate-high, high), insurance type (Medicaid, self-employed, employee), Body Mass Index (BMI), Charlson Comorbidity Index (CCI), and year (2009 to 2014).

### 2.3. Ethical Consideration

This study was approved by the Institutional Review Board, Eulji University (IRB number: EUIRB2019-13).

### 2.4. Patient and Public Involvement

Patients and or the public were not involved in this study. There are no plans to disseminate the research results to study participants.

### 2.5. Statistical Analysis

The distribution of each categorical variable was examined by an analysis of frequencies and percentages, and χ^2^ tests were performed. *T*-tests were also performed for continuous variables to compare mean and standard deviation values. In the fully adjusted model, all variables were entered simultaneously. The generalized estimating equation (GEE) model was used to identify these variables and the incidence of CVD while controlling for potential confounding variables. Cox proportional hazard modeling was performed that included both patient characteristics and detailed CVD onset. The start date was defined as the date of initial diagnosis of dyslipidemia or prescribed medication, and the last date was the date of CVD diagnosis or the end of the study periods (31 December 2015) or death date. We used the Poisson regression model to evaluate associations between regions and the average number of outpatient visits. The gamma generalized linear model, using the log link function, was used to evaluate healthcare expenditure differences between regions. All statistical analyses were performed using SAS statistical software version 9.4 (SAS Institute, Cary, NC, USA). A *p*-value < 0.05 was considered statistically significant. Additionally, we used the Statistical Geographic Information Service by Statistics Korea to create regional distribution maps for dyslipidemia and the development of the cardiovascular disease.

## 3. Results

The general characteristics of the study population are shown in Table 1. A total of 52,377 patients were newly diagnosed with dyslipidemia, and 12,443 of these (23.8%) were diagnosed with CVD. Regions with the lowest populations (<100,000) had the highest CVD risk (*n* = 1635; 28.8%), and regions with more than 200,000 people had similar CVD risks (22.2–23.1%). Patients with diabetes (*n* = 3832; 30.9%) had a higher risk of CVD than patients without diabetes (*n* = 8611; 21.5%). Patients with high-risk scores also had the highest risk for CVD (scores = 0, 16.6%; scores = 1, 27.9%; scores = 2, 38.4%; scores >3, 53.7%). Average outpatient visits (mean ± SD) were higher for patients who had CVD (4.18 ± 3.08) than for non-CVD patients (2.93 ± 2.58, *p* < 0.0001). Similar to the results for outpatient visits, healthcare costs were also higher for CVD patients (KRW 83,887 ± 133,275) than for non-CVD patients without (KRW 64,262 ± 153,921, *p* < 0.0001).

Figure 1 shows the regional population classifications, the number and distribution of dyslipidemia patients per 100,000 population by region, and the distribution of CVD in dyslipidemia patients. In general, the population was concentrated in the capital and metropolitan areas, and dyslipidemia was more prevalent in rural areas where the population density was lowest. The incidence of CVD in dyslipidemia patients was higher in regions with the lowest population densities.

Figure 2 shows the distribution of abnormal serum cholesterol levels by region. In Korea, dyslipidemia patients showed a higher proportion (31.3%) of abnormalities in TC levels, and a lower proportion (12.5%) of abnormalities for low HDL-C levels. In regions of low population density, the proportion of abnormal TG levels was highest, but the proportion of high LDL-C levels was lowest. In regions with populations of more than 200,000, high TC had a similar distribution, and the proportion of low HDL-C was lower in regions with populations over 400,000.

Table 2 shows the association between the region and the risk of CVD. CVD risk increased in low-density regions compared to high-density regions, but only regions with populations less than 100,000 were statistically significant (odds ratio [OR], 1.147; 95% confidence interval [CI], 1.051–1.252). CVD risk was higher for patients with diabetes than for patients without diabetes (OR, 1.070; 95% CI, 1.017–1.125). Higher risk scores were significantly associated with an increase in CVD (score 1: OR: 1.460, 95% CI: 1.387–1.536; score 2: OR: 2.035, 95% CI: 1.898–2.182; score ≥ 3: OR: 3.591, 95% CI: 3.010–4.283). Of the CVD types, the risk of IHD was higher in regions with population less than 300,000 compare to those in high-density region (<100,000: hazard ratio (HR): 1.137 95% CI: 1.146–1.561; <300,000: HR: 1.263, 95% CI: 1.100–1.450). Similar results were observed for cerebrovascular disease, with increased risk in small population densities.

Table 3 shows the results of the GEE model for the association between region and healthcare expenditure and outpatient visiting. Healthcare expenditures were higher in low-density areas compared to the high-density areas, but this difference was only significant in areas with less than 100,000 people (Rate Ratio [RR], 1.072; 95% CI, 1.017–1.130). Outpatient visits were also higher in low-density regions compared to high-density regions. Healthcare expenditures (RR, 1.461; 95% CI, 1.409–1.515) and outpatient visits (RR, 1.620; 95% CI, 1.592–1.646) were higher for patients with diabetes than for patients without diabetes.

## 4. Discussion

Health disparities within a population are major concerns in many countries, and efforts have been made to bridge these disparity gaps for those affected. Such disparities also exist in Korea, and health inequalities are increasing not only by socio-economic status but also by region [20,21]. This study aimed to evaluate regional disparities in the dyslipidemia prevalence, health utilization, and the risk of CVD.

In general, the prevalence of dyslipidemia was higher in low population-density regions compared to those with high population densities. The most frequent findings for dyslipidemia were high TC, LDL-C, and TG. These results are similar to those of previous Korean studies, and confirm the differences seen between Korea and other countries that have high TG and low HDL levels [22,23,24]. Dyslipidemia distributions varied within *si-gun-gu* areas of the same district. Possible explanations for differences in serum cholesterol distributions may be related to regional variations in socio-economic status, dietary habits, physical activity [6,15], and differences in the quality of available healthcare.

We also found that the risk of CVD was highest in regions with low population densities, especially in those with populations under 100,000, compared to regions with high population densities. These results may also be related to differences in the quality of healthcare between regions. Access to care is one of the most important factors for preventing disease and having better patient care outcomes [25]. In general, most of the large hospitals in Korea are concentrated in the capital area, rather than in less-populated rural areas, so high-quality healthcare services are only available in regions with high population densities. Therefore, there may be quality gaps between regions, and compared to patients living in urban areas, rural dyslipidemia patients may receive relatively lower quality healthcare and have worse outcomes [3,26]. In these vulnerable areas, the role of primary care providers will be important. In clinical practice, primary healthcare providers should educate patients with dyslipidemia to take their medications regularly to prevent the risk of CVD. In particular, patients with diabetes or at high risk of CVD will need early intensive intervention or management, such as regular exercise and change diet habits, to reduce the risk of CVD, and the role of the primary healthcare provider will be important. Finally, dyslipidemia patients with low socioeconomic status are at high risk for CVD, and these patients should be properly managed through social support along with regular health examination for their health status.

Healthcare utilization, assessed by healthcare expenditures and outpatient visits, was higher in low population-density regions than in those with higher population densities. Patients in rural areas may visit more hospitals than urban patients, leading to increased healthcare expenditures, and despite more visits to healthcare providers, patients from low-density areas did not have better results than those from high-density areas. These results provide evidence that a regional approach is needed for reducing gaps in both healthcare and patient health.

Recently, the Korean government introduced a pilot program for community-based healthcare that provides comprehensive care at the community level for patients with chronic diseases. This approach underlines the importance of healthcare at the community level, and regional differences should also be considered for successful policymaking. This study provides evidence that there are regional differences in the quality of healthcare as well as the prevalence of the chronic disease. To reduce these differences in quality and disease burden, a region-based approach should be considered, especially for quality improvement in low-density areas. More research is needed to clarify regional differences in population health and healthcare quality.

This study has several strengths. First, we used data from a large representative cohort sample, so the results should be considered meaningful for policymakers. Second, although there have been many previous reports on regional disparities in healthcare, no research exists for regional disparities in healthcare utilization and patient outcomes. This study provides evidence for regional healthcare disparities and highlights the importance of a regional approach to reduce quality gaps between regions. Third, the results suggest that Asian patients with dyslipidemia differ from those in Western countries, a significant finding in any health-gap study.

Despite these strengths, this study does have some limitations. First, patient factors that were not considered in our studies, such as physical activity behavior, level of education, dietary habits, and occupation, may have influenced CVD risk. Second, we did not consider physician-related factors that might affect patient outcomes. Patient outcomes can vary depending on a healthcare provider’s ability to manage chronic diseases. Third, regions were classified only according to the population. It is possible that within this classification, other factors besides population density could have explained the results depending on the region (e.g., Seoul, as the capital city). However, we do not consider this potential bias to be large because we used a nested model that accounted for district regions to reduce inter-regional variation. Finally, it is possible that other, unmeasured factors may have affected these quality gaps between regions, and further research is needed to take these factors into account.

## 5. Conclusions

This study examined regional variations in dyslipidemia prevalence, healthcare utilization, and the risk of CVD. Regional prevalence variations occurred according to population density, with low-density regions having a higher risk for CVD, more visits to healthcare providers, and more healthcare spending. To bridge these regional health gaps, consideration should be given not only to general socio-economic factors, but also to specific regional factors, and a region-based approach should be adopted. Finally, healthcare providers should be considered early intensive intervention or management to reduce the risk of CVD in patients living in a vulnerable region.

## Figures and Tables

**Figure 1 ijerph-18-00538-f001:**
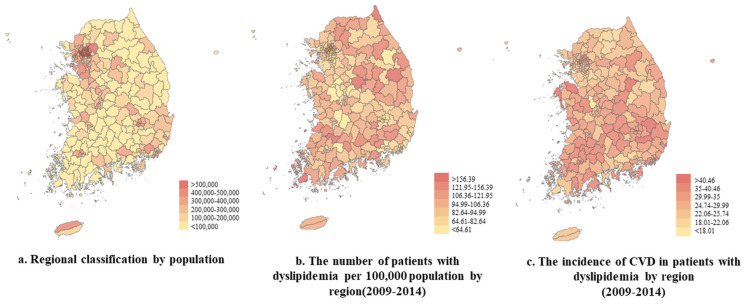
Regional classification by population, and the distributions of dyslipidemia and CVD by region (2009–2014).

**Figure 2 ijerph-18-00538-f002:**
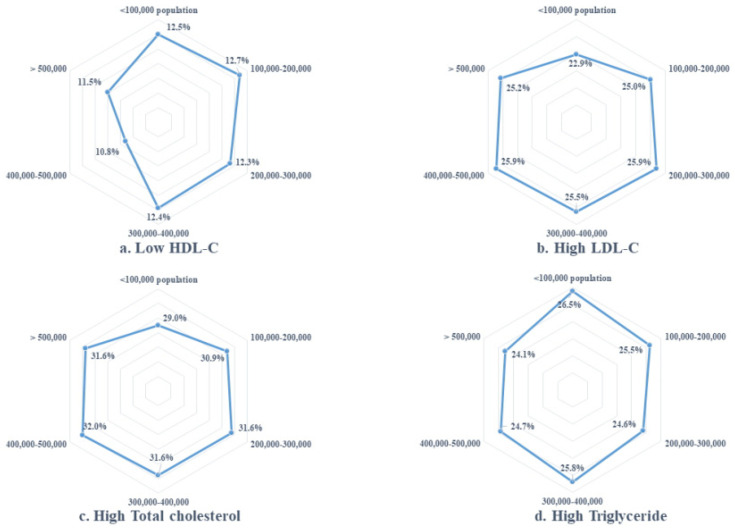
Regional disparities for abnormal serum cholesterol levels. The percentages represent patients with high (TC: ≥240 mg/dL, TG: ≥200 mg/dL, LDL-C: ≥160 mg/dL) or low (HDL-C: <40 mg/dL) serum cholesterol levels.

**Table 1 ijerph-18-00538-t001:** Baseline characteristics of dyslipidemia patients and their association with CVD (*n* = 52,377) (Unit: N/M, %/SD).

Variables	CVD						*p*-Value
	Yes		No		Total		
Region							
<100,000 population	1635	(28.8)	4039	(71.2)	5674	(10.8)	<0.0001
100,000–200,000	1674	(24.8)	5085	(75.2)	6759	(12.9)	
200,000–300,000	3042	(23.1)	10,120	(76.9)	13,162	(25.1)	
300,000–400,000	2517	(23.1)	8405	(77.0)	10,922	(20.9)	
400,000–500,000	2095	(22.2)	7357	(77.8)	9452	(18.1)	
>500,000	1480	(23.1)	4928	(76.9)	6408	(12.2)	
Diabetes							
Yes	3832	(30.9)	8556	(69.1)	12,388	(23.7)	<0.0001
No	8611	(21.5)	31,378	(78.5)	39,989	(76.4)	
Prescribed Medication							
Yes	1955	(17.4)	9257	(82.6)	11,212	(21.4)	<0.0001
No	10,488	(25.5)	30,677	(74.5)	41,165	(78.6)	
Risk score							
0	4363	(16.6)	21,955	(83.4)	26,318	(50.3)	<0.0001
1	5374	(27.9)	13,884	(72.1)	19,258	(36.8)	
2	2370	(38.4)	3805	(61.6)	6175	(11.8)	
≥3	336	(53.7)	290	(46.3)	626	(1.2)	
CCI	2.29	±2.55	1.73	±2.11	1.86	±2.23	<0.0001
BMI	24.83	±3.23	24.07	±3.13	24.25	±3.17	<0.0001
Healthcare utilization(per year)							
Costs (Unit: KRW)	83,887	±133,275	64,262	±153,921	68,924	±149,507	<0.0001
outpatient visiting	4.18	± 3.08	2.93	± 2.58	3.22	± 2.76	<0.0001
Sex							
Male	5813	(24.1)	18,308	(75.9)	24,121	(46.1)	0.0906
Female	6630	(23.5)	21,626	(76.5)	28,256	(54.0)	
Age							
20–34	265	(7.7)	3184	(92.3)	3449	(6.6)	<.0001
35–49	2795	(16.7)	13,984	(83.3)	16,779	(32.0)	
50–64	6492	(26.0)	18,502	(74.0)	24,994	(47.7)	
≥65	2891	(40.4)	4264	(59.6)	7155	(13.7)	
Income							
Low	2999	(24.2)	9379	(75.8)	12,378	(23.6)	0.1396
Low-moderate	2989	(23.3)	9865	(76.8)	12,854	(24.5)	
Moderate-high	2754	(23.4)	9042	(76.7)	11,796	(22.5)	
High	3701	(24.1)	11,648	(75.9)	15,349	(29.3)	
Insurance							
Medicaid	325	(28.7)	806	(71.3)	1131	(2.2)	<0.0001
Self-Employed	4173	(25.2)	12,375	(74.8)	16,548	(31.6)	
Employees	7945	(22.9)	26,753	(77.1)	34,698	(66.3)	
Year of diagnosis							
2009	2146	(33.8)	4208	(66.2)	6354	(12.1)	<0.0001
2010	3866	(34.1)	7487	(66.0)	11,353	(21.7)	
2011	2348	(25.7)	6798	(74.3)	9146	(17.5)	
2012	1747	(20.2)	6898	(79.8)	8645	(16.5)	
2013	1313	(16.0)	6919	(84.1)	8232	(15.7)	
2014	1023	(11.8)	7624	(88.2)	8647	(16.5)	
Total	12,443	(23.8)	39,934	(76.2)	52,377	(100.0)	

CVD: cardiovascular disease; CCI: Charlson comorbidity index; 1$ = 1090.3 KRW, adjusted for gross price inflation- that is, as if the gross-to-cost ratio had stayed constant since 2009.

**Table 2 ijerph-18-00538-t002:** Regional association between CVD risk.

Variables	CVD			Types of CVD								
				Ischemic Heart Disease			Cerebrovascular Disease			Hypertension		
	OR	95% CI		HR	95% CI		HR	95% CI		HR	95% CI	
Region												
<100,000 population	1.147	1.051	1.252	1.137	1.146	1.561	1.181	1.008	1.384	1.056	0.973	1.146
100,000–200,000	1.055	0.969	1.149	1.169	1.000	1.366	1.105	0.941	1.297	0.995	0.918	1.079
200,000–300,000	1.041	0.966	1.122	1.263	1.100	1.450	1.172	1.018	1.350	0.976	0.909	1.049
300,000–400,000	1.016	0.940	1.098	1.105	0.956	1.276	1.008	0.868	1.171	0.998	0.927	1.074
400,000–500,000	0.970	0.895	1.051	1.111	0.958	1.289	1.048	0.900	1.221	0.930	0.862	1.005
>500,000	1.000	-	-	1.000	-	-	1.000	-	-	1.000	-	-
Diabetes												
Yes	1.070	1.017	1.125	1.008	0.924	1.010	1.063	0.972	1.163	1.058	1.009	1.109
No	1.000	-	-	1.000	-	-	1.000	-	-	1.000	-	-
Prescribed Medication												
Yes	0.730	0.689	0.774	0.712	0.637	0.796	0.774	0.692	0.867	0.732	0.691	0.776
No	1.000	-	-	1.000	-	-	1.000	-	-	1.000	-	-
Risk score												
0	1.000	-	-	1.000	-	-	1.000	-	-	1.000	-	-
1	1.460	1.387	1.536	1.336	1.219	1.465	1.205	1.095	1.326	1.440	1.368	1.516
2	2.035	1.898	2.182	1.553	1.377	1.751	1.348	1.190	1.528	2.003	1.878	2.137
≥3	3.591	3.010	4.283	2.022	1.545	2.648	1.808	1.385	2.362	3.129	2.745	3.567
CCI	1.066	1.055	1.077	1.094	1.076	1.112	1.126	1.108	1.145	1.034	1.024	1.044
BMI	1.082	1.074	1.089	1.032	1.019	1.045	1.007	0.993	1.021	1.086	1.079	1.093
Sex												
Male	1.025	0.978	1.075	1.254	1.155	1.361	1.099	1.009	1.196	0.987	0.944	1.033
Female	1.000	-	-	1.000	-	-	1.000	-	-	1.000	-	-
Age												
20–34	1.000	-	-	1.000	-	-	1.000	-	-	1.000	-	-
35–49	2.285	1.997	2.613	2.941	2.106	4.106	3.348	2.132	5.259	2.044	1.782	2.345
50–64	3.623	3.169	4.143	5.151	3.700	7.172	7.756	4.964	12.117	2.821	2.460	3.234
≥65	6.195	5.364	7.155	7.028	4.989	9.899	15.884	10.094	24.994	4.326	3.744	4.999
Income												
Low	1.082	1.017	1.150	0.918	0.824	1.022	0.891	0.795	0.998	1.195	1.127	1.267
Low-moderate	1.085	1.023	1.151	0.930	0.839	1.031	0.962	0.863	1.071	1.166	1.102	1.234
Moderate-high	1.037	0.976	1.102	0.919	0.829	1.020	0.931	0.835	1.038	1.082	1.021	1.147
High	1.000	-	-	1.000	-	-	1.000	-	-	1.000	-	-
Insurance												
Medicaid	1.212	1.047	1.403	1.229	0.959	1.575	1.733	1.375	2.186	1.041	0.908	1.192
Self-Employed	1.044	0.997	1.094	1.054	0.972	1.143	1.035	0.951	1.127	1.055	1.010	1.102
Employees	1.000	-	-	1.000	-	-	1.000	-	-	1.000	-	-
Year of diagnosis												
2009	1.000	-	-	1.000	-	-	1.000	-	-	1.000	-	-
2010	0.829	0.770	0.892	0.803	0.708	0.910	0.779	0.680	0.892	1.036	0.967	1.109
2011	0.600	0.555	0.648	0.721	0.629	0.825	0.704	0.608	0.816	0.956	0.887	1.029
2012	0.437	0.403	0.474	0.587	0.505	0.682	0.632	0.540	0.741	0.868	0.801	0.941
2013	0.327	0.300	0.357	0.580	0.493	0.683	0.691	0.583	0.819	0.835	0.765	0.911
2014	0.218	0.199	0.239	0.544	0.457	0.649	0.555	0.458	0.673	0.870	0.792	0.956

OR: Odds ratio; HR: Hazard Ratio; 95% CI: confidence interval; CCI: Charlson Comorbidity Index.

**Table 3 ijerph-18-00538-t003:** The association between region and healthcare expenditure and outpatient visiting.

Variables	Healthcare Expenditure			Outpatient Visiting		
	RR	95% CI		RR	95% CI	
Region						
<100,000 population	1.072	1.017	1.130	1.080	1.050	1.111
100,000–200,000	1.048	0.971	1.131	1.039	1.012	1.067
200,000–300,000	1.041	0.994	1.090	1.024	1.001	1.047
300,000–400,000	1.050	0.989	1.115	1.026	1.002	1.050
400,000–500,000	1.019	0.974	1.066	1.030	1.005	1.055
>500,000	1.000	-	-	1.000	-	-
Diabetes before dyslipidemia						
Yes	1.461	1.409	1.515	1.620	1.595	1.646
No	1.000	-	-	1.000	-	-
Prescribed Medication						
Yes	0.702	0.680	0.725	1.018	1.002	1.035
No	1.000	-	-	1.000	-	-
Risk score for CVD						
0	1.000	-	-	1.000	-	-
1	1.028	0.980	1.078	1.043	1.026	1.059
2	0.982	0.940	1.025	1.064	1.039	1.089
≥3	1.030	0.943	1.124	1.101	1.038	1.168
CCI	1.064	1.055	1.073	1.033	1.029	1.036
BMI	1.009	1.000	1.017	1.013	1.011	1.015
Sex						
Male	0.912	0.880	0.945	0.887	0.874	0.901
Female	1.000	-	-	1.000	-	-
Age						
20–34	1.000	-	-	1.000	-	-
35–49	1.071	0.945	1.215	1.340	1.300	1.382
50–64	1.119	0.988	1.267	1.549	1.502	1.598
≥65	1.145	0.997	1.314	1.648	1.589	1.710
Income						
Low	0.988	0.930	1.049	1.047	1.027	1.067
Low-moderate	0.975	0.935	1.017	1.038	1.018	1.057
Moderate-high	1.018	0.966	1.072	1.036	1.017	1.055
High	1.000	-	-	1.000	-	-
Insurance						
Medicaid	1.395	1.265	1.539	1.244	1.178	1.313
Self-Employed	1.023	0.977	1.072	1.013	0.998	1.028
Employees	1.000	-	-	1.000	-	-
Year of diagnosis						
2009	1.000	-	-	1.000	-	-
2010	1.259	1.164	1.361	1.065	1.039	1.092
2011	1.112	1.046	1.183	0.983	0.958	1.009
2012	0.996	0.933	1.063	0.959	0.934	0.985
2013	0.983	0.914	1.057	0.927	0.903	0.953
2014	0.971	0.911	1.034	0.929	0.904	0.955

RR: Rate Ratio/95% CI: confidence interval.

## Data Availability

The authors have no authority over the data, and the data is provided upon request to the National Health Insurance Services.

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
