# Peer review of "Regional Prevalence of Dyslipidemia, Healthcare Utilization, and Cardiovascular Disease Risk in South Korean: A Retrospective Cohort Study"

_ijerph, 2021, doi:10.3390/ijerph18020538_

Round 1

Reviewer 1 Report

Thank you for the opportunity to review this study that explores the topic: Regional Prevalence of Dyslipidemia, Healthcare Utilization, and Cardiovascular Disease Risk in South Korean Patients with Dyslipidemia: A Retrospective Cohort Study. Below is some feedback intended to help the authors strengthen the manuscript.

Title

The title refers twice to "dyslipidemia". Authors should improve the title so that this repetition does not happen.

Abstract

The Abstract clearly presents the background, the purpose of the study and the methodology. The main results are presented, as well as the main conclusions of the study.

Introduction

The authors in the introduction make a correct review of the literature using a wide variety of studies carried out in the thematic area of the present study. The referred studies are current and relevant to the topic. At the end of the introduction, the authors present the purpose of the study and some generic options. I suggest that these options go to the method section.

Material and Methods

The materials and methods include the important information to understand how the data were obtained and how the statistical analysis was carried out. However, ethical considerations in accessing and processing data should be better described.

Results

Table 1 does not identify the risk assessment area. I suggest that the percentage values presented are not per category within each variable but are general for the total variable. BMI must be separated by a line. I suggested that age be categorized. In my opinion, this table should be divided into sociodemographic and clinical variables.

Table 2 does not identify the risk assessment area. In the tables I suggest that the most evident results be highlighted in bold.

Discussion

The discussion of the results is very interesting. I just suggest that the authors could discuss further the implications for clinical practice and the measures that can be implemented to decrease the dyslipidemia prevalence.

Conclusions

The authors present the main conclusions resulting from the study and the main measures to be implemented. The authors must further develop the implications for clinical practice. That is, what can be improved by knowing the results of this study.

Author Response

Revision Note for ijerph-1049007

Title: Regional prevalence of dyslipidemia, healthcare utilization, and cardiovascular disease risk in South Korean patients with dyslipidemia: A retrospective cohort study

First, we greatly appreciate the comments and suggestions offered by the reviewers, which we used to improve the manuscript. Our response to each comment follows, and we have attached a revision note and also highlighted the revised sections of the manuscript. Again, thank you for the valuable and helpful comments.I have attached the response file for reviewer #1.

Reviewer 2 Report

Authors presented a valuable, interesting, well designed study conducted on a large cohort of patients, thus making it highly representative. Its results may impact future policy-making or serve as a baseline for further assessment of the effectiveness of current healthcare policies. 

I think it would be useful to use it as a starting point to investigate other factors that might impact the regional differences in prevalence of CVD risk factors and healthcare utilization, such as level of health awareness, physical activity, dietary habits. The authors mention some of these potential factors in their discussion, so I think they are well aware of the significance of their findings and the directions in which they want to pursue their further studies. 

Statistical methods used were appropriate. 

It's a good quality study, concisely presented, definitely worthy of publication. 

Author Response

First, we greatly appreciate the comments and suggestions offered by the reviewers, which we used to improve the manuscript. Our response to each comment follows, and we have attached a revision note and also highlighted the revised sections of the manuscript. Again, thank you for the valuable and helpful comments.

Answer to Reviewer 2:

I think it would be useful to use it as a starting point to investigate other factors that might impact the regional differences in prevalence of CVD risk factors and healthcare utilization, such as level of health awareness, physical activity, dietary habits. The authors mention some of these potential factors in their discussion, so I think they are well aware of the significance of their findings and the directions in which they want to pursue their further studies. 

Answer: Thank you for your comments. We revised our manuscript as following(page 1 line 35-36):

Cardiovascular disease (CVD) is the leading cause of death worldwide, and many factors have been identified as targets for reducing its prevalence. Most population CVD risk factors, such as high total cholesterol levels and hypertension, are modifiable and can be reduced by changes in behavior [1,2]. In other words, changes in healthcare and adoption of healthy behaviors, including physical activity and dietary habit, in the general population mean that people can live healthier lives. However, some populations are more vulnerable than others due to health and healthcare disparities, and these health gaps between different populations have been recognized as an important issue to be addressed [3].

Reviewer 3 Report

Lines 54- 55- Is there a citation to support the statement that "dyslipidemia is considered less important than other chronic diseases such as hypertension and diabetes", or is this the opinion of the authors?

Line 129- Should this read "Patients and/or 'the' public.."?

Line 165- Should this read "dyslipidemia 'was' more prevalent..."?

line 251- Good example of dietary habits as a patient factor. Other patient factors to consider include physical activity behavior, level of education, and occupation.

Overall, this is an interesting, relevant, and insightful study.

Author Response

First, we greatly appreciate the comments and suggestions offered by the reviewers, which we used to improve the manuscript. Our response to each comment follows, and we have attached a revision note and also highlighted the revised sections of the manuscript. Again, thank you for the valuable and helpful comments.

Answer to Reviewer 3:

Lines 54- 55- Is there a citation to support the statement that "dyslipidemia is considered less important than other chronic diseases such as hypertension and diabetes", or is this the opinion of the authors?
Answer: Thank you for your comments.

Previous study suggested that ‘Dyslipidemia is becoming increasingly common although most middle-aged Koreans are not aware of this condition leading to low control rate. ’. Similar results were observed in statistics in Korea, the prevalence of dyslipidemia in 2017 is 21.5%, but the awareness rate is 58.9% and the treatment rate was less than 50%, whereas awareness rate of diabetes and hypertension was 70%. Patient’s perception of disease is related to the importance of disease, and people tend to be well aware of disease that are considered relatively important. I have written a sentence by referring to previous research.

Line 129- Should this read "Patients and/or 'the' public.."?
Answer: Thank you for your comments. We revised it per your comments(page 3 line 128).

Line 165- Should this read "dyslipidemia 'was' more prevalent..."?
Answer: Thank you for your comments. We revised it per your comments(page 5 line 164).

line 251- Good example of dietary habits as a patient factor. Other patient factors to consider include physical activity behavior, level of education, and occupation.
Answer: Thank you for your comments. We revised it per your comments(page 9 line 253-254).